# Macro-Modeling for N-Type Feedback Field-Effect Transistor for Circuit Simulation

**DOI:** 10.3390/mi12101174

**Published:** 2021-09-29

**Authors:** Jong Hyeok Oh, Yun Seop Yu

**Affiliations:** ICT & Robotics Engineering and IITC, Hankyong National University, 327 Jungang-ro, Anseong-si 17579, Gyenggi-do, Korea; rnjsdlr7@hknu.ac.kr

**Keywords:** feedback field-effect transistor, macro-model, compact modeling, hybrid inverter, spike neural network

## Abstract

In this study, we propose an improved macro-model of an N-type feedback field-effect transistor (NFBFET) and compare it with a previous macro-model for circuit simulation. The macro-model of the NFBFET is configured into two parts. One is a charge integrator circuit and the other is a current generator circuit. The charge integrator circuit consisted of one N-type metal-oxide-semiconductor field-effect transistor (NMOSFET), one capacitor, and one resistor. This circuit implements the charging characteristics of NFBFET, which occur in the channel region. For the previous model, the current generator circuit consisted of one ideal switch and one resistor. The previous current generator circuit could implement *I_DS_*-*V_GS_* characteristics but could not accurately implement *I_DS_*-*V_DS_* characteristics. To solve this problem, we connected a physics-based diode model with an ideal switch in series to the current generator circuit. The parameters of the NMOSFET and diode used in this proposed model were fitted from TCAD data of the NFBFET, divided into two parts. The proposed model implements not only the *I_DS_*-*V_GS_* characteristics but also the *I_DS_*-*V_DS_* characteristics. A hybrid inverter and an integrate and fire (I&F) circuit for a spiking neural network, which consisted of NMOSFETs and an NFBFET, were simulated using the circuit simulator to verify a validation of the proposed NFBFET macro-model.

## 1. Introduction

Recently, many problems occurring in designing next-generation integrated circuits (ICs) have been emerging, owing to continuous scaling for a metal-oxide-semiconductor field-effect transistor (MOSFET). The nanoscale MOSFET has a short channel effect, which increases the standby power by increasing leakage current, and has difficulty in controlling threshold voltage owing to the gate field influence becoming weak in a nanoscale channel [1,2]. Moreover, using MOSFETs to design high-performance ICs is difficult due to the physical limitations of MOSFETs being unable to have a subthreshold swing (SS) of less than 60 mV/dec at room temperature. To overcome these problems, various next-generation devices are suggested [3,4,5,6,7,8]. Among them, the family of Z^2^-FET (or field-effect diode) and feedback field-effect transistor (FBFET) with band-modulation operating mechanisms are in the spotlight for substituting the MOSFET [8,9,10,11,12]. The FBFET works on positive feedback in the channel region that is divided into two regions. One is a gated region and the other is an ungated region. These two regions form a potential barrier and potential well, respectively. The potential barrier and potential well, controlled by gate-source voltage and drain-source voltage, inject carriers into the potential well. Subsequently, positive feedback occurs in the channel region. The energy band of the channel region changes abruptly from S-shape to parallel, owing to the positive feedback. These characteristics of the FBFET cause a high on/off current ratio and a sharp increase and decrease in the forward sweep and the reverse sweep, respectively. Furthermore, the FBFET has hysteresis characteristics that are caused by a difference in the threshold voltage of the forward and reverse sweeps. Various applications of the FBFET utilizing these characteristics have been studied. The electrical characteristics of the FBFET have been investigated using the FBFET as a logic gate [10,11,12,13,14,15,16]. In addition, memory circuits, neuromorphic circuits, and biosensors consisting of FBFETs or hybrid components (e.g., the FBFET and the MOSFET) were investigated utilizing charging in the channel region and the hysteresis characteristics [17,18,19,20,21,22].

The various applications of the FBFET can be analyzed using technology computer-aided design (TCAD) [23]. TCAD mixed-mode [23] is a powerful tool for investigating the electrical characteristics of the various types of circuit consisting of FBFETs. However, to analyze and simulate the very large-scaled ICs consisting of FBFETs or hybrid components, a compact model of FBFET for circuit simulation is needed. Thus far, compact models of FBFET have been studied with physics-based analytical equations [24,25,26], and yet, only one macro-model based on an equivalent circuit consisting of only Simulation Program with Integrated Circuit Emphasis (SPICE) devices has been reported. The reported macro-model of FBFET consists of two circuits [27]. One is a charge integration circuit and the other is a current generation circuit. The charge integration circuit is composed of one MOSFET and one capacitor, and the current generation circuit consists of one ideal switch and one resistor. This model has a limitation on implementing the *I_DS_*-*V_DS_* characteristics of FBFET. Therefore, there is a problem in simulating the memory circuits or neuromorphic circuits with SPICE using that model. To simulate the circuits consisting of the FBFETs or hybrid components, solutions to overcome these limitations are required.

In this paper, we propose an improved FBFET macro-model that can implement not only *I_DS_*-*V_GS_* characteristics but also *I_DS_*-*V_DS_* characteristics by adding one physics-based diode model to the model presented previously by another group. First, the electrical characteristics and the macro-model configuration of the FBFET are discussed in Section 2. Thereafter, the parameter extraction of the MOSFET and diode used in this proposed model is described, and SPICE simulation results of the proposed macro-model are discussed in Section 3. Finally, the conclusions of this study are discussed in Section 4.

## 2. Macro-Modeling of FBFET

### 2.1. FBFET Mechanism

Figure 1a shows a 2D schematic diagram of an N-type FBFET (NFBFET). Figure 1b shows the energy band diagrams of the NFBFET at off states (*V_GS_* = *V_DS_* = 0 V). Figure 1c,d shows the energy band diagrams at *V_DS_* = 1 V and different *V_GS_*, and at *V_GS_* = 0.6 V and different *V_DS_*, respectively. In Figure 1c,d, the red and black lines denote on and off states of the NFBFET, respectively, and the symbols e^-^ and h^+^ represent electron and hole, respectively. Moreover, the black dotted lines denote energy band of near-threshold voltage of the forward sweep. The FBFET structure is a p-i-n diode structure (Z^2^-FET) [28] with a high-doped n-p diode in the channel region. As the drain-source voltage (*V_DS_*) or the gate-source voltage (*V_GS_*) are not applied, the drain-source region forms potential barriers, as shown in Figure 1b. Therefore, the carriers are not injected into the channel region. To make the electrons (holes) flow from (to) the source to (from) the drain, appropriate *V**_DS_* and *V_GS_* must be applied to eliminate potential barriers. The energy band diagrams, as shown in Figure 1c,d, represent the carriers’ flow over the threshold of *V_GS_* and *V_DS_* = 1 V, as well as over the threshold of *V_DS_* and *V_GS_* = 0.6 V, respectively. In Figure 1c, when *V_GS_* is applied at *V_DS_* = 1 V, the potential barrier (gated channel region) is lower. The electron from the source region is injected into the potential well (ungated channel region). Thereafter, the accumulated electrons make the potential well higher. The hole, which cannot flow into the channel region by the potential well, can be injected into the channel region. This positive feedback rapidly eliminates the potential barrier and potential well. In the end of the forward sweep, the energy band of the NFBFET is aligned, as denoted by the red line in Figure 1c. In Figure 1d, when *V_DS_* is applied at *V_GS_* = 0.6 V, the electrons from the source region are injected into the channel region (ungated channel region). The positive feedback occurs with the same mechanism mentioned in Figure 1c. By these characteristics, the NFBFET has a high on/off ratio. Moreover, the NFBFET has hysteresis characteristics, owing to differences between the forward and reverse sweeps.

### 2.2. TCAD Simulation

Table 1 shows the structure parameters of the NFBFET. The total length of the NFBFET is 140 nm. The oxide material used in the simulation is aluminum oxide (Al_2_O_3_). To simulate the NFBFET, the MOSFET and bipolar junction transistor models were used. Those models include the Lombardi concentration, voltage, and temperature model (CVT), Shockley–Read–Hall model (SRH), Fermi–Dirac calculation, field-dependent mobility model (FLDMOB), Auger recombination model (AUGER), and band gap narrowing model (BGN). The simulation temperature was 300 K, and the TCAD simulator Atlas by Silvaco was used for simulation [23]. Figure 2a,b shows the TCAD simulation results of the drain-source currents to gate-source voltage (*I_DS_*-*V_GS_*) characteristics at *V_DS_* = 1 V and drain-source currents to drain-source voltage (*I_DS_*-*V_DS_*) characteristics at *V_GS_* = 0.6 V of NFBFET. Voltage directions of the forward and reverse sweeps are indicated by the dashed arrows in Figure 2. The NFBFET has a memory window that is the difference in threshold voltage between forward and reverse sweeps. The memory windows of the NFBFET are 0.53 V and 2.06 V for the *I_DS_*-*V_GS_* characteristics and *I_DS_*-*V_DS_* characteristics, respectively.

### 2.3. Macro-Model of NFBFET

Figure 3a,b shows the equivalent circuits of the previous macro-model [27] and the proposed macro-model of NFBFET, respectively. Both NFBFET macro-models consist of two parts. One is a charge integration circuit consisting of one voltage source, one N-type MOSFET (NMOSFET), one capacitor, and one resistor. The other is a current generation circuit consisting of one resistor and one ideal switch. The charge integration circuit implements the charging characteristics of charges in the channel region of the FBFET. The charges accumulated on the potential well in the channel region are integrated into the capacitor *C_body_*. The current generation circuit implements generating the on/off current by switching the ideal switch. In the previous model (Figure 3a), the current generation circuit works by using only an ideal switch with a simple diode equation. Since the simple diode equation in the previous model is used, it is difficult to exactly fit *I_DS_*-*V_DS_* characteristics and the on-current of FBFET by using only one parameter *I_off_*. This is valid only for the *I_DS_*-*V_GS_* characteristics. However, for the *I_DS_*-*V_DS_* characteristics, the NFBFET has diode characteristics in reverse sweep, as shown in Figure 2b. To implement the *I_DS_*-*V_DS_* characteristics more precisely, the physics-based diode model (red dotted circle) embedded in SPICE, as shown in Figure 3b, is connected at the ideal switch in series for the proposed model. For the SPICE simulation using the proposed macro-model of NFBFET, the Berkeley Short-channel IGFET Model - Independent Multi-Gate (BSIM-IMG, level = 301, version 102.7.0) [29,30] model was used for the MOSFET, the diode (level = 3) model was used for diode model, and the SmartSpice program was used for the simulation [31].

## 3. SPICE Simulation Results

### 3.1. Parameter Fitting

The NFBFET was divided into two parts for the parameter fitting. One is a p-n junction from the drain region for the diode, and the other is an n-p-n structure from the source region for the NMOSFET, as shown in Figure 1a. This concept came from a geometric view of the NFBFET. The parameters of the BSIM-IMG model for the NMOSFET and the diode (level = 3) model were first fitted with the TCAD data of the NFBFET. The parameters of the BSIM-IMG model and diode model were extracted, as shown in Table 2. In the equivalent circuit, the values of other components are *C_body_* = 1 pF, *R_body_* = 450 Ω, *V_T_* = 0.000610945 V, *V_H_* = 0.000290705 V, *R_off_* = 1/*I_off_* = 1/(6.496 × 10^−15^) Ω, and *R_on_* = 1 Ω.

Figure 4a shows the results of the diode parameter fitting. The square symbols and black lines denote the simulation results by TCAD and SPICE, respectively. There were mismatches at a high field (>1.3 V) for linear plot and at a low field (~0.4 V) for log plot. These were caused by difficulties in fitting the parameter of a heavily doped diode. Figure 4b,c shows *I_DS_*-*V_GS_* characteristics and *I_DS_*-*V_DS_* characteristics of NMOSFET in NFBFET, respectively. The square symbols and solid lines denote the simulation results by TCAD and SPICE, respectively. Likewise, there were mismatches for both results. These mismatches were caused by fitting the heavily doped channel of MOSFET, since the BSIM-IMG model was limited to below 5 × 10^18^ cm^−3^ in terms of the channel doping concentration [29,30].

### 3.2. SPICE Results of Macro-Model

Figure 5a,b shows *I_DS_*-*V_GS_* characteristics and *I_DS_*-*V_DS_* characteristics of the NFBFET, respectively. The symbols and lines denote the simulation results by TCAD and macro-model of SPICE, respectively. For the *I_DS_*-*V_GS_* characteristics, both results at *V_GS_* = 1 V and *V_GS_* = 0.8 V were fitted with TCAD data in terms of threshold voltages. However, for the subthreshold region and *V_DS_* below 0.8 V, there were mismatches. In particular, for the reverse sweep, the threshold voltage mismatches were observed at *V_DS_* below 0.8 V. The *I_DS_*-*V_GS_* characteristics are implemented by an ideal switch in the current generation circuit. The ideal switch only made an on/off current; therefore, these characteristics caused mismatches near the threshold voltage and at the subthreshold region, as shown in Figure 5a. For the *I_DS_*-*V_DS_* characteristics, the hysteresis characteristics occur under specific conditions, which are controlled by sustaining drain voltage with continuity of diffusion and generation-recombination currents in the channel region [28]. The hysteresis characteristics only occur near *V_GS_* = 0.6 V, as shown in Figure 5b. Likewise, there were mismatches in the subthreshold region and on-current as the BSIM-IMG and diode model used for the macro-model have a limitation in terms of heavily doped channel, as described in Section 3.1.

### 3.3. Model Validation

Figure 6 shows *I_DS_*-*V_DS_* characteristics of NFBFET. The squares and solid lines denote the experimental data and fitted SPICE results, respectively. The experimental results are from the Z^2^-FET structure with 28 nm fully depleted silicon-on insulator (FDSOI) fabrication [16]. Although this NFBFET structure is not the same as the one in Figure 1a, the proposed macro-model of NFBFET can fit the electrical characteristics for the same operating mechanism, as shown in Figure 6. The parameters of the proposed macro-model of NFBFET were changed to *V_T_* = 0.00032591 V and *V_H_* = 0.00032591 V for the ideal switch model and RS = 1 × 10^−6^ Ω·m^2^ for the diode model. The small mismatches in the parameter fitting result occurred due to the limitation of the models, as described in Section 3.1.

Figure 7a,b shows a circuit diagram of a hybrid inverter, which consists of P-type MOSFET (PMOSFET) and NFBFET, as shown in Figure 1a, and the voltage transfer characteristics (VTC) of the hybrid inverter, respectively. The square symbols and solid lines denote VTCs simulated with TCAD and macro-model of SPICE, respectively, as shown in Figure 7b. The VTC by TCAD has a switching voltage gap caused by the threshold voltage difference between forward and reverse sweeps. The VTC for the macro-model also has a switching voltage gap, but the switching slope is extremely sharp compared with the TCAD results. These characteristics were caused by the hysteresis voltage control of the ideal switch model composed in the current generation circuit.

Figure 8a shows the circuit diagram of an integrate and fire (I&F) circuit for a spiking neural network (SNN) [21]. Complementary MOS (CMOS)-based I&F circuits have a number of MOSFETs for implementing spike and reset mechanisms; therefore the circuits have a high power consumption. To overcome this problem, the other research team suggested an I&F circuit that consisted of the NFBFET. The NFBFET has the capability to accumulate charge in the channel region. Using these characteristics, the spike and reset mechanism can be implemented. Moreover, the NFBFET replaces a number of transistors, which significantly reduces system area and power consumption. Figure 8b show the results of SPICE simulation for an I&F circuit consisting of a macro-model of NFBFET, as shown in Figure 1a. The black line, red line, and square symbols denote synaptic current (*I_synaptic_*), membrane voltage (*V_MB_*), and spike voltage (*V_spike_*), respectively. Word line 1 (*V_WL1_*) and word line 2 (*V_WL2_*) voltages were applied at 0.6 V and 0.1 V, respectively. The spike operation occurred for the I&F consisting of a macro-model, but the complete reset operation was not performed, as shown in Figure 8b. This was due to the magnitude of hysteresis characteristics of the macro-model.

## 4. Conclusions

In this paper, we proposed an improved macro-model of an NFBFET compared with a previous model. The previous model was configured with a charge integration circuit and a current generation circuit. The current generation circuit of the previous model has a limitation that cannot implement *I_DS_*-*V_DS_* characteristics. To solve this problem, we added a diode into the current generation circuit. First, for simulating the NFBFET with the proposed macro-model using SPICE, the parameters of the NMOSFET and diode composed in the model were fitted with the simulation results of the NFBFET and diode by TCAD. The fitted results of the diode and NMOSFET had mismatches in the subthreshold region and high field region. These mismatches were caused by a p-n-p-n structure with only a heavily doped region (10^20^ cm^−3^). The parameters of the NMOSFET and diode in the macro-model of NFBFET were extracted from the simulation results by TCAD. The SPICE simulation results of the macro-model of NFBFET had mismatches near the threshold voltage and at the subthreshold region. These mismatches were caused by a limitation that implemented transfer characteristics using the diode and ideal switch. Finally, for validating the macro-model of the NFBFET, a parameter fitting was conducted with experimental data, and a hybrid inverter and an I&F circuit, which included a MOSFET and a NFBFET, were simulated by SPICE. The fitted results using the parameters extracted from the experimental data show that the macro-model of the NFBFET can implement the electrical characteristics for the same operating mechanism. The VTC of TCAD and macro-model had a switching voltage gap, but the switching slope of the macro-model was extremely sharp, owing to an ideal switch in the current generation circuit. The I&F circuit generated spike operation, but the complete reset operation was not performed due to the limitation of hysteresis characteristics. To overcome the limitations such as the mismatch around the threshold voltages of the macro-model of NFBFET consisting of MOSFET and the other devices, improvements in the proposed macro-model or the analytical modeling of FBFET are required. Moreover, since the proposed structure may include leakage currents such as band-to-band tunneling (BTBT), an improved macro-model development and parameter extractions are required.

## Figures and Tables

**Figure 1 micromachines-12-01174-f001:**
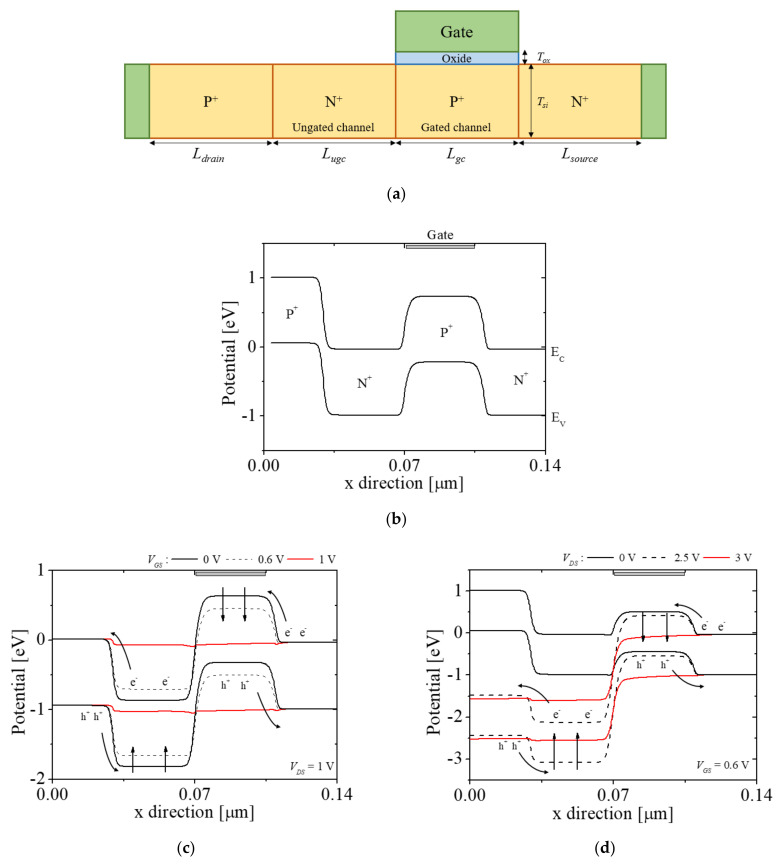
(**a**) Schematic diagram of an N–type feedback field–effect transistor (NFBFET). The energy band diagrams of the NFBFET; (**b**) at off state (*V_DS_* = 0 V and *V_GS_* = 0 V); (**c**) at *V_DS_* = 1 V and different *V_GS_*; and (**d**) at *V_GS_* = 0.6 V and different *V_DS_*.

**Figure 2 micromachines-12-01174-f002:**
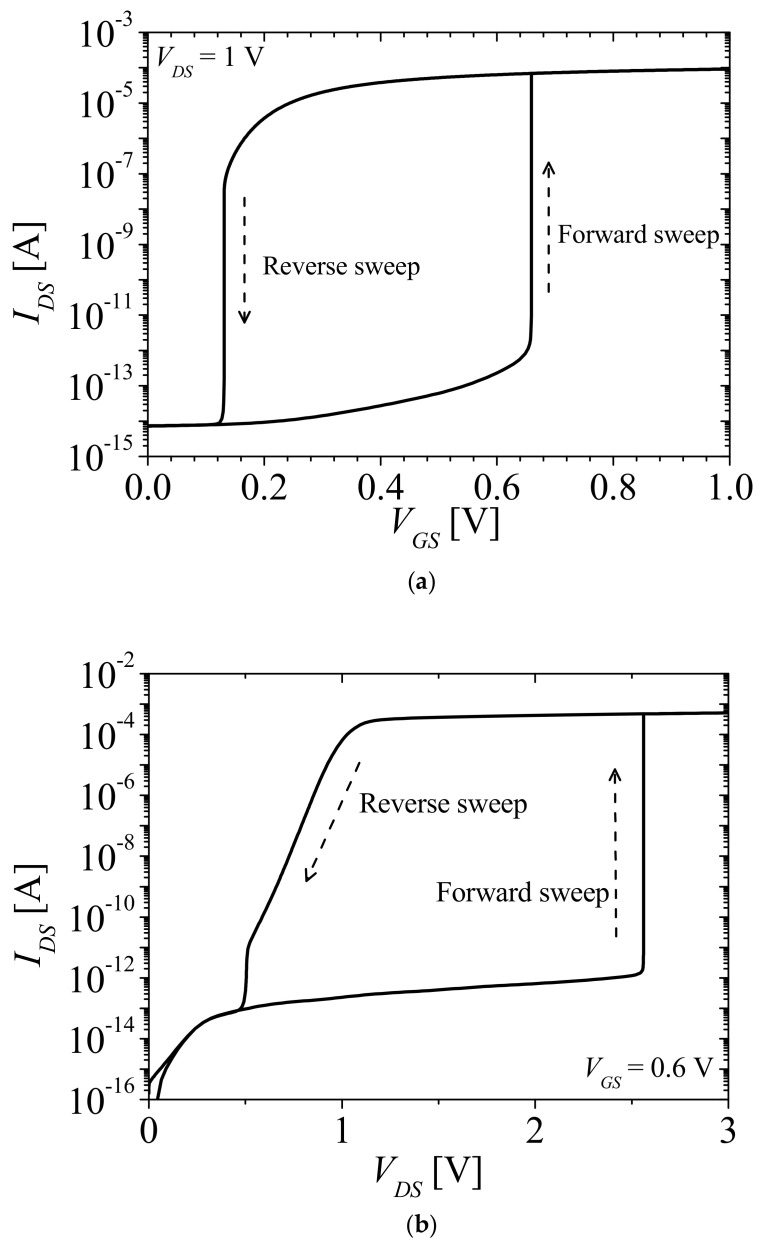
(**a**) *I_DS_**-V_GS_* characteristics at *V_DS_* = 1 V; (**b**) *I_DS_**-V_DS_* characteristics at *V_GS_* = 0.6 V.

**Figure 3 micromachines-12-01174-f003:**
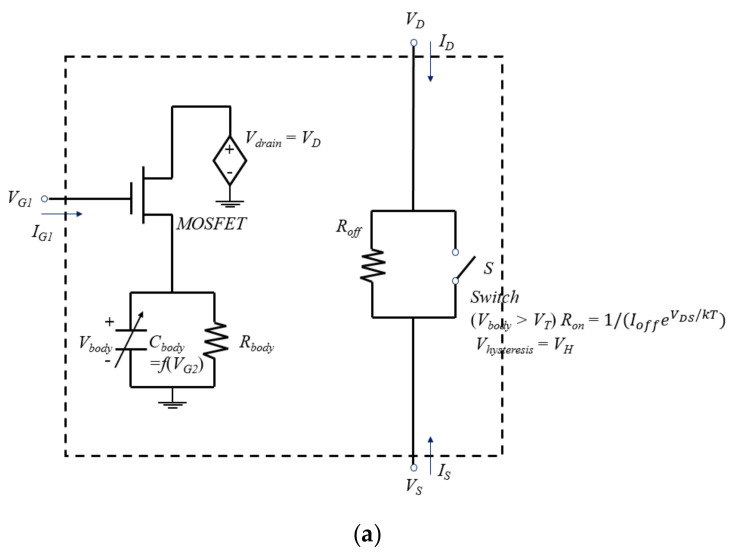
Circuit diagram of (**a**) the previous macro-model and (**b**) the proposed macro-model.

**Figure 4 micromachines-12-01174-f004:**
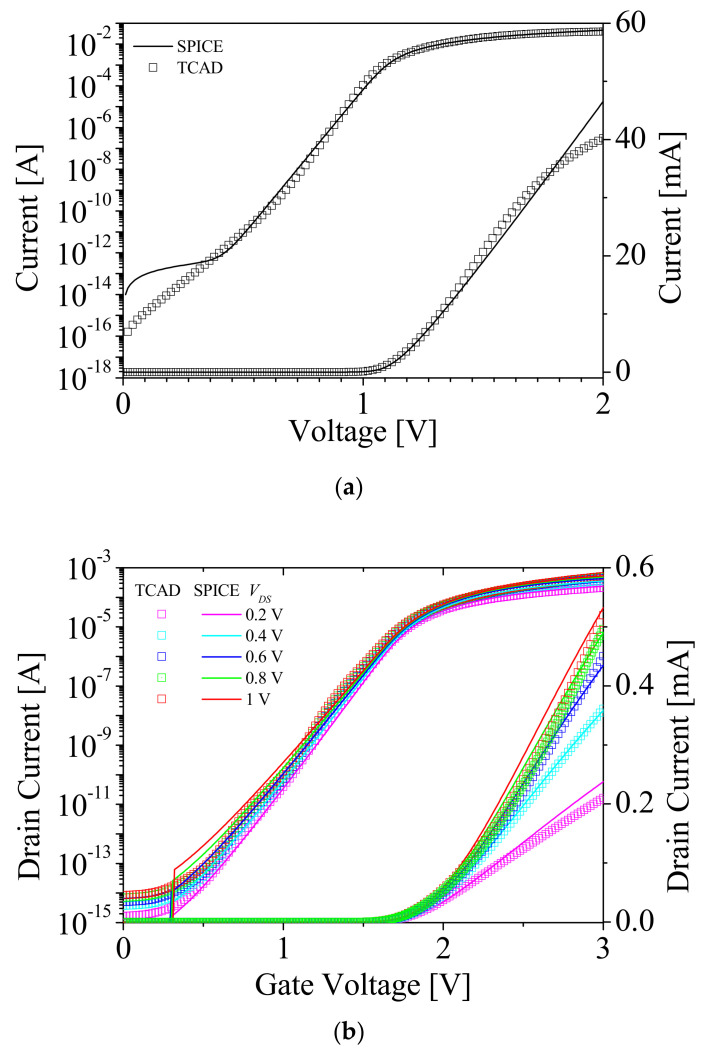
Diode parameter fitting data (**a**) and metal–oxide–semiconductor field–effect transistor (MOSFET) parameter fitting data for (**b**) *I_DS_*-*V_GS_* characteristics, and (**c**) *I_DS_*-*V_DS_* characteristics.

**Figure 5 micromachines-12-01174-f005:**
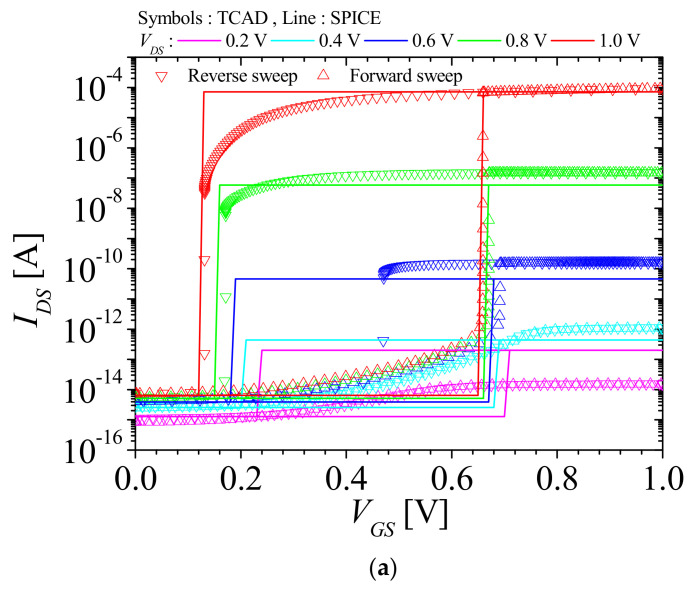
Simulation Program with Integrated Circuit Emphasis (SPICE) results of macro–model of NFBFET for (**a**) *I_DS_*-*V_GS_* characteristics and (**b**) *I_DS_*-*V_DS_* characteristics.

**Figure 6 micromachines-12-01174-f006:**
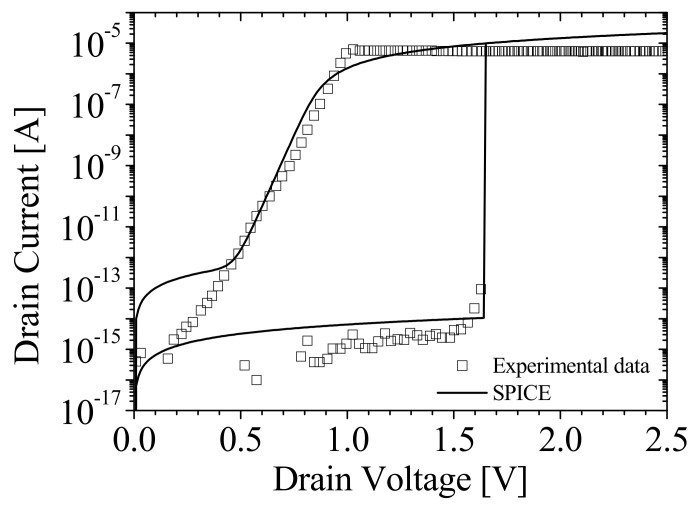
*I_DS_*–*V_DS_* characteristics: Symbols and lines denote experimental data and simulation results with the parameter extracted from experimental data [16], respectively.

**Figure 7 micromachines-12-01174-f007:**
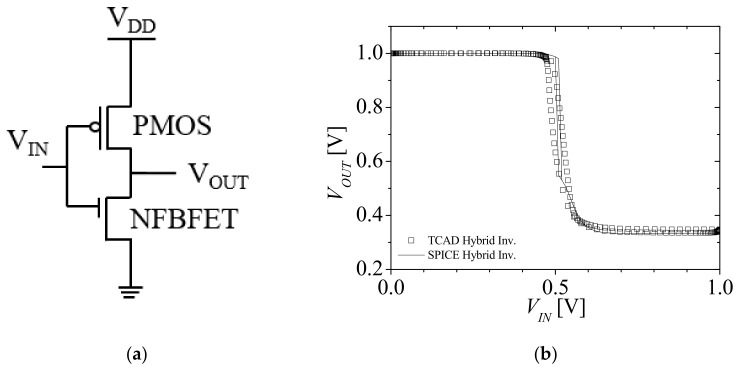
Circuit diagram of (**a**) a hybrid inverter and (**b**) the voltage transfer characteristics (VTCs) for TCAD and SPICE.

**Figure 8 micromachines-12-01174-f008:**
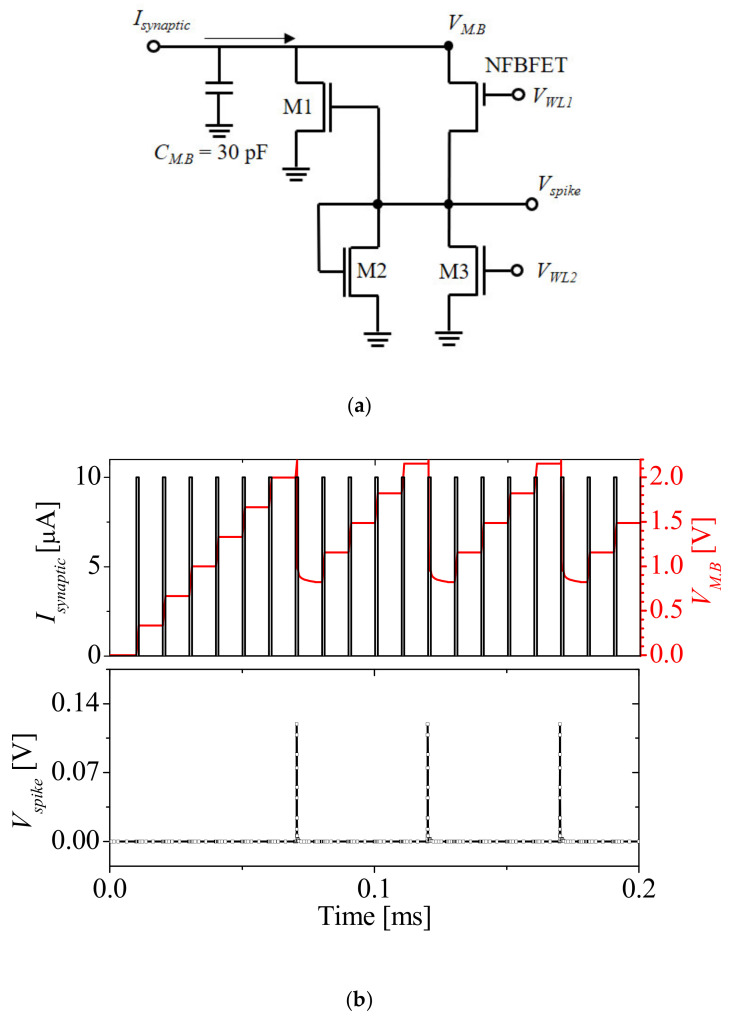
Circuit diagram of (**a**) an integrate and fire circuit for spiking neural network (SNN) and (**b**) synaptic current, membrane voltage, and spike voltage.

**Table 1 micromachines-12-01174-t001:** Structure parameters of the NFBFET used in technology computer-aided design (TCAD) simulation.

Parameters	Description	Value/Unit
*L_drain_*	Length of drain region	30 nm
*L_gc_*	Length of gated channel region/P^+^ region	40 nm
*L_ugc_*	Length of ungated channel region/N^+^ region	40 nm
*L_source_*	Length of source region	30 nm
*T_si_*	Thickness of silicon body	15 nm
*T_ox_*	Thickness of oxide	3 nm
	P^+^ region doping concentration	$1\times 10^{20}{\;\mathrm{cm}}^{-3}$
	N^+^ region doping concentration	$1\times 10^{20}{\;\mathrm{cm}}^{-3}$
*Φ_N_*	Gate work function of N-type FBFET	4.6 eV

**Table 2 micromachines-12-01174-t002:** Summary of parameters extraction of Berkeley Short–channel IGFET Model – Independent Multi–Gate (BISM–IMG) and diode model (Level = 3).

Parameters	Unit	Description	Value
BSIM-IMG model (Level = 301)			
L	m	Gate length	40 × 10^−9^
W	m	Gate width	1 × 10^−6^
DVT0	-	SCE coefficient	29.2
DVT1	-	SCE exponent coefficient	1.0
VSAT	m/s	Saturation velocity in the saturation region	1 × 10^4^
AVSAT	-	Saturation velocity in the saturation region for short channel devices	4 × 10^3^
VSAT1	m/s	Saturation velocity in the linear region	5 × 10^4^
AVSAT1	-	Saturation velocity in the linear region for short channel devices	4 × 10^5^
PTWG	1/V^2^	Correction factor for velocity	2.8
U0	m^2^/V·s	Low field mobility	5 × 10^−3^
PCLM	-	Channel length modulation (CLM) parameter	0.035
APCLM	-	Channel length modulation (CLM) parameter for short channel devices	0.98
RDSW	Ω·μ_m_	Zero bias S/D extension resistance per unit width	0.1
PDIBL1	-	Parameter for DIBL effect on Rout	0.9
PDIBL2	-	Parameter for DIBL effect on Rout	2 × 10^−4^
Diode model (Level = 3)			
L	m	Length of diode	70 × 10^−9^
W	m	Width of diode	1 × 10^−6^
IS	A/m^2^	Saturation current	1 × 10^−12^
ISW	A/m^2^	Sidewall saturation current	1 × 10^−14^
RS	Ω·m^2^	Ohmic resistance	3 × 10^−11^
N	-	Emission coefficient	1.08

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
