# Peer review of "Macro-Modeling for N-Type Feedback Field-Effect Transistor for Circuit Simulation"

_micromachines, 2021, doi:10.3390/mi12101174_

Round 1

Reviewer 1 Report

see attached

Author Response

Please find a file of the answer to your comments.

Reviewer 2 Report

The manuscript (micromachines-1359378), Macro-modeling for N-type Feedback Field Effect Transistor for Circuit Simulation, shows the results of the FEFET device applications and modeling. Authors presented comprehensive analysis in this work. However, some technical comments still need to be addressed before further confirm the outcomes, shown as following - 

  1. For the mechanism presentation, please provide step-by-step schematic plots with the applied voltage conditions. Since the Vds is 1V, the schematic diagram did not show the Vds energy potential difference in the plot. Please clarify the mechanism in detail by several plots and mark which conditions now for the Vds and Vgs conditions.
  2. Can authors provide the experimental data to confirm the FBFET really works? How does the switching speed in FBFET as compared to the CMOS? Also, how to confirm that the electrons in the N++ region would not be "leaky" (by tunneling effect) in the P-N junctions (remember, the N+ and P+ distance is quite short)? Have authors added the tunneling modeling in TCAD simulation or any related devices have been fabricated for the characterizations? This referee is quite curious about the performance in this device structure as compared to conventional one. Please provide a simple benchmark Table to to simulate the performance difference in these two. 
  3. Can PMOS work in FBFET? What is the energy band diagram? Please also compare it to the conventional PMOS device. 

Due to the above comments, this referee would like to put the manuscript status as "Major Revision" in the current phase. 

Author Response

(The authors gave the same response as above.)

Reviewer 3 Report

In this work, Oh et al proposed a macro-model of an N-type feedback field effect transistor (NFBFET) with improved functionality. By adding one physics-based diode model, the model proposed in this work can implement both IDS-VGS characteristics and IDS-VDS characteristics. The authors provided electrical characteristics and the macro-model configuration of the feedback field effect transistors. Simulation results were shown and discussed. This work provided some useful insights for the feedback field-effect transistor simulation. I recommend publishing of this work after some minor points below addressed.

1. Line 14 on page 1, the authors missed "switch" after "ideal".
2. Remove all the red wave lines from a couple of figures (Figure 1a, Figure 3).
3. For the SPICE simulation result, how the mismatch around threshold voltages can be improved with other circuits simulation strategy?

Author Response

(The authors gave the same response as above.)

Round 2

Reviewer 1 Report

see attached

Author Response

Please find an attached file for the answers to Reviewer1's comments.

Yun Seop Yu

Reviewer 2 Report

Authors have replied to this referee' comments in detail. No further comments from this referee. 

Author Response

Thanks for your kind comments.

Round 3

Reviewer 1 Report

Fig 2c does not exist

nV precision in Vt,h is ridiculous